# Genetic Diversity and Signatures of Selection in a Native Italian Horse Breed Based on SNP Data

**DOI:** 10.3390/ani10061005

**Published:** 2020-06-08

**Authors:** Michela Ablondi, Christos Dadousis, Matteo Vasini, Susanne Eriksson, Sofia Mikko, Alberto Sabbioni

**Affiliations:** 1Dipartimento di Scienze Medico-Veterinarie, University of Parma Via del Taglio 10, 43126 Parma, Italy; christos.dadousis@unipr.it (C.D.); alberto.sabbioni@unipr.it (A.S.); 2Libro Genealogico Cavallo Bardigiano, Associazione Regionale Allevatori dell’Emilia-Romagna, 43126 Parma, Italy; matteo.vasini@araer.it; 3Department of Animal Breeding and Genetics, Swedish University of Agricultural Sciences, PO Box 7023, S-75007 Uppsala, Sweden; susanne.eriksson@slu.se (S.E.); sofia.mikko@slu.se (S.M.)

**Keywords:** conservation, selection scan, autochthonous, genomics, coat color, *LCORL*, IBH, ROH, *MC1R*, horse

## Abstract

**Simple Summary:**

The Bardigiano horse is a native Italian breed bred for living in rural areas, traditionally used in agriculture. The breed counts about 3000 horses, and it is nowadays mainly used for recreational purposes. The relatively small size and the closed status of the breed raise the issue of monitoring genetic diversity. We therefore characterized the breed’s genetic diversity based on molecular data. We showed a critical reduction of genetic variability mainly driven by past bottlenecks. We also highlighted homozygous genomic regions that might be the outcome of directional selection in recent years, in line with the conversion of Bardigiano horses from agricultural to riding purposes.

**Abstract:**

Horses are nowadays mainly used for sport and leisure activities, and several local breeds, traditionally used in agriculture, have been exposed to a dramatic loss in population size and genetic diversity. The loss of genetic diversity negatively impacts individual fitness and reduces the potential long-term survivability of a breed. Recent advances in molecular biology and bioinformatics have allowed researchers to explore biodiversity one step further. This study aimed to evaluate the loss of genetic variability and identify genomic regions under selection pressure in the Bardigiano breed based on GGP Equine70k SNP data. The effective population size based on Linkage Disequilibrium (N_e_) was equal to 39 horses, and it showed a decline over time. The average inbreeding based on runs of homozygosity (ROH) was equal to 0.17 (SD = 0.03). The majority of the ROH were relatively short (91% were ≤2 Mbp long), highlighting the occurrence of older inbreeding, rather than a more recent occurrence. A total of eight ROH islands, shared among more than 70% of the Bardigiano horses, were found. Four of them mapped to known quantitative trait loci related to morphological traits (e.g., body size and coat color) and disease susceptibility. This study provided the first genome-wide scan of genetic diversity and selection signatures in an Italian native horse breed.

## 1. Introduction

The Bardigiano is an Italian autochthonous horse breed whose origin can be traced back to the age of the Roman Empire in the Belgian Gaul province [1]. The breed consists of a homogeneous population characterized by typical and distinct traits: (a) the height ranges between 140 and 149 cm for males, and 135 and 147 cm for females; (b) the only admitted coat color is bay, with a preference for dark bay, while chestnuts and extremely light bays are not allowed; (c) limited white markings on the legs and face are allowed, although not preferred. Conformation related characteristics include a small head with a straight or concave profile, low withers, a slightly straight back, deep girth and an overall muscular appearance [2]. The rustic and docile Bardigiano horse was traditionally used as a working horse in mountain areas, and for meat production. After the Second World War, the number of Bardigiano horses dramatically decreased, numbering only five stallions and 150 mares [3]. The decrease in population size led to the creation of the Bardigiano studbook in 1977, and since then the Bardigiano has been considered a purebred breed due to the closure policy applied by the studbook. Nowadays, the Bardigiano breed counts roughly 3000 live horses, mainly used for riding and light draft purposes. It is classified as a small native population, possibly at risk of extinction [4]. However, the relatively small size and the closed status of the current population raises the issue of monitoring genetic diversity. A recent study exploring genetic diversity in the Bardigiano breed at pedigree level highlighted alarming levels of inbreeding and a loss of genetic diversity [5]. In this study, we present an extension of the above-mentioned work by using genomic data to analyze the genetic diversity in the breed.

Recent techniques in genomics permit present researchers to gain more insights into genetic diversity, population history and selection signatures compared to pedigree-based methods [6,7]. Several genome-wide population structure and genetic diversity studies have been performed in livestock species [8,9,10,11]. The first comprehensive insight into equine genetic diversity among a large breed cohort was published by Petersen et al. in 2013 [12]. This study used fixation index (Fst) statistics, one of the most popular methods for the capture of between-breed divergence [9,13,14,15]. Runs of homozygosity (ROH) are likewise a well-established method, widely used to detect the within-breed loss of genetic diversity [16]. Runs of homozygosity are long consecutive homozygous segments distributed across the genome. Among other evolutionary forces, they also arise from identical-by-descendent haplotypes which came from common ancestors [17]. Therefore, ROHs have been used nowadays as a valuable source of information for the description of genomic inbreeding (F_ROH_) [18,19,20,21]. Compared to pedigree-based inbreeding, F_ROH_ is able to capture variation due to Mendelian sampling and linkage during gamete formation [7]. In addition, F_ROH_ does not rely on pedigree quality, which may be a limiting factor in inbreeding estimation based on genealogical data [22,23]. Overlapping homozygous regions, highly shared among individuals belonging to the same population, are called ROH islands. Since directional artificial selection reduces genomic variability, ROH islands are thought to be potential signs of selection around a target locus [16,24,25]. Several recent examples of ROH and population structure analyses applied to European horse breeds show key aspects of history and selection pressure [25,26,27,28,29,30,31,32,33]. However, few studies were conducted in the framework of European small native horse breeds [29,31,32,34] and, to the best of our knowledge, none of them specifically analyzed Italian autochthonous horse breeds. Therefore, based on a medium-density single nucleotide polymorphism (SNP) genotype panel, we aimed to investigate genomic diversity in the Bardigiano breed as an example of the reservoir of Italian native breeds.

The detailed aims of this study were to (i) investigate genomic effective population size throughout generations, (ii) calculate within-breed genomic diversity based on F_ROH_, and (iii) evaluate the presence of potential signs of selection based on ROH islands.

## 2. Materials and Methods

### 2.1. Sample Collection and Genotyping

The study included genotype data from 89 Bardigiano horses (38 males and 51 females). The horses submitted to genotyping were carefully selected to represent the most of genetic variability from the pedigree information [5]. The following criteria were applied in the selection procedure: (i) verification of pedigree depth, excluding animals with an equivalent complete generation (defined as the sum for (1/2)^n^, where n is the number of generations separating the individual from each of its known ancestors [35]) lower than two; (ii) horses with available conformation measurements and linear and traditional evaluations were preferred [2]; (iii) no more than three animals descending from the same stallion were allowed; (iv) animals had to belong to the last two generations, considering an average generation interval of 8.74 years [5]. All samples were genotyped with the GGP Equine70k^®^ (Illumina, San Diego, CA, USA) SNP chip, containing 65,157 SNPs separated on average by 40 kb. The SNP physical positions were remapped from the former reference genome EquCab2 to EquCab3 [36], as described in [37]. Only SNPs located on the 31 *Equus caballus* autosome (ECA) chromosomes were retrieved and used in this study. The quality control (QC) was performed in PLINK v1.9 [38] by removing SNPs with a call rate lower than 0.90, a Hardy–Weinberg equilibrium (HWE) deviation with *p* < 10^−6^ and minor allele frequencies (MAF) <0.01. Horses with a call rate lower than 0.90 were excluded.

### 2.2. Effective Population Size

Past and present effective population sizes (N_e_) were estimated with the SNeP v.1.1 software [39] using only horses in the last birth year cohort (2011–2019). SNeP v.1.1 estimates the trends of the historical effective population size trajectories from SNP data based on linkage disequilibrium (LD). The recombination rate was calculated using the Sved and Feldman’s mutation rate modifier [40], and sample size correction was made for unphased genotypes. The cM unit in this study was set to 1.24 Mbp [41]. The minimum and maximum distance between pairs of SNPs was set to 0.5 Mbp and 26 Mbp, respectively. We also performed an N_e_ slope analysis (N_e_S) [42] to look into the rate and directionality of N_e_ changes occurring throughout generations. The N_e_S analysis identifies subtle changes in the inferred N_e_ curve not visually explicit in the N_e_ plot. The slope of each segment linking pairs of neighboring N_e_ estimates was first calculated and then normalized using the median of the two most proximal past N_e_ slope values.

### 2.3. Runs of Homozygosity

The ROH segments were detected using the DetectRUNS [43] package in R [44], and defined as follows: (i) at least 15 SNPs in a run, (ii) a minimum length of a run equal to 500 kb, (iii) a maximum distance between consecutive SNPs in a window 1000 kb, (iv) a lower density limit of 1 SNP per 100 kb [27] and (v) allowing for a maximum of one missing and one heterozygous SNP in a run [25]. The ROH segments were divided into the following five length classes: 0.5–1 Mbp, 1–2 Mbp, 2–4 Mbp, 4–8 Mbp and >8 Mbp. The total number of ROHs (N_ROH_), average length of ROHs (L_ROH_) and the average ROH number (S_ROH_) were summarized according to each ROH length category. The genomic inbreeding coefficients (F_ROH_) were calculated following the method described in [18]:(1)$$F_{ROH}=\Sigma \frac{L_{ROH}}{L_{AUTO}}$$
with *L_ROH_* being the length of ROHs in each individual and *L_AUTO_* being the length of the autosomal genome, which was set to 2276 Gb, based on the genome length covered by SNPs. Based on the hypothesis that the length of ROH reflects the chronological time points when inbreeding happened [20], the genomic inbreeding was expressed separately for five length ROH categories (0.5–1 Mbp, 1–2 Mbp, 2–4 Mbp, 4–8 Mbp, >8 Mbp). By using the formula 1/2 g Morgan, with g being equal to generation [20,45], and the relationship between Centimorgan and Mb in horses [41], we estimated the timepoint of the inbreeding event based on ROH length. The inbreeding was also calculated per chromosome, and the type of distribution was evaluated based on skewness and kurtosis.

Putative ROH islands were determined based on overlapping homozygous regions within more than 70% of the horses. The EqCab3 genomic coordinates of these regions were used to retrieve candidate gene lists and annotations from the Biomart web interface in Ensembl [46] and from the UCSC genome browser platform [47]. Putative ROH islands were compared with quantitative trait loci (QTL) regions previously identified and reported in the Horse QTL database [48]. Functional analyses were performed on the genes potentially under selection, located in ROH shared in more than 85% of the animals [25] or overlapping with known QTLs.

## 3. Results

### 3.1. Genotyping Quality Control

All horses except one passed the genotype data quality control (37 males, 51 females), and 58,047 SNPs were retrieved. The average genotype call rate was 0.99 and the average pedigree depth based on complete generation equivalent of the selected horses was 6.28 (SD = 0.79). The horses descended from 54 stallions with an average of 1.65 offspring each (SD = 0.82) and 76 mares with 1–2 offspring each. In total, 80% of the horses were born between 2011 and 2019, whereas 18 horses were born between 2001 and 2010.

### 3.2. Effective Population Size

For the N_e_ calculation, we only included animals born between 2011 and 2019, which numbered 70 horses. The N_e_ declined over time, and was estimated to be ~39 horses one generation ago. Instead, the estimated N_e_ 20 generations ago was around 195 horses (Figure 1).

To investigate the change in slope of the inferred N_e_ obtained from LD-based estimation, we used the NeS method, which offers more detailed information about population changes 1–20 generations ago. A constant rate of change is shown as a flat line at 0 in the Y-axis, whereas deviations above and below 0 represent relative increases and reductions, respectively, of the N_e_ change in slope compared to the previous two generations. This analysis highlighted two sharp reductions in N_e_ in the Bardigiano breed, being approximately ten and six generations ago (Figure 2).

### 3.3. Runs of Homozygosity

A total of 28,423 ROHs were found among the 88 Bardigiano horses analyzed in this study. An average of 323 ROHs were found per horse, with a maximum number equal to 365 and minimum of 285 ROHs. The average length of ROHs per individual was equal to 1215.8 Kb ± 173.7 Kb (SD). The majority of ROHs were shorter than 2 Mbp, with 69.5% being shorter than 1 Mbp and 21.4% being between 1 and 2 Mbp length (Table 1). The proportion of ROHs longer than 8 Mbp was equal to 1.2%, with an average length of 13.4 Mbp. A total of 84 horses exhibited ROHs in the >8 Mbp length class, with an average of 4.2 ROHs per horse in this class and a minimum and maximum equal to 1 and 13 ROHs, respectively.

The number of ROHs and length of ROH segments varied across chromosomes, as shown in Figure 3. The highest number of ROHs was identified on ECA1 (2386), while the lowest was identified on ECA31, with 273 ROHs detected. Nevertheless, when normalizing for the chromosome length, the highest portion was found on ECA29 and ECA28. The highest chromosome length covered by ROH was detected on ECA10, with 1.49 Mbp covered by ROHs, followed by ECA5 (1.46 Mbp), ECA1 (1.43 Mbp) and ECA3 (1.40 Mbp).

#### 3.3.1. ROH as a Measure of Inbreeding

The inbreeding, measured as the proportion of the genome covered by ROH, resulted in an average F_ROH_ equal to 0.17 ± 0.03 (SD). To differentiate ancient and recent inbreeding, we calculated F_ROH_ based upon five ROH length classes, as presented in Table 2. Inbreeding events occurring over ~31 generations ago were presented by F_ROH_ calculated per length below or equal to 2 Mbp. For the ROH length class above 8 Mbp, four horses did not have ROHs and thus their inbreeding resulted in being equal to 0. The degree of inbreeding based on the longest ROH class showed an average inbreeding equal to 0.02, reflecting inbreeding events which happened <8 generations ago.

The inbreeding calculated per chromosome showed large variation among chromosomes both in terms of average values and the distribution of inbreeding level per individual (Figure 4). The highest average inbreeding was found in ECA 1 (mean = 0.21 ± 0.09 SD) and the lowest on ECA 30 (0.12 ± 0.09 SD). For all chromosomes except ECA 3, ECA 15 and ECA 17, a highly skewed distribution towards higher values was found (i.e., a distribution skewness above one). A kurtosis value above three was used to investigate the presence of outlier individuals per each chromosome. In total, 21 out of the 31 chromosomes showed a kurtosis higher than three, highlighting the presence of some horses exhibiting an excess of homozygosity.

#### 3.3.2. ROH as a Measure to Detect Signatures of Selection

A total of five chromosomes (ECA3, ECA10, ECA11, ECA15 and ECA19) contained ROH islands that were shared by more than 70% of individuals in the total sample. Three ROH islands were detected on ECA3, two on ECA10, and the remainder were equally distributed among ECA11, ECA15 and ECA19 (Figure 5). The three ROH islands located and ECA3 were shared in more than 85% of the animals, and in particular the ROH island located on ECA3 (ECA3: 35.48–36.01 Mbp) was shared in 93% of the animals (82 out of the 88).

Table 3 reports the genomic coordinates of the ROH islands and the annotated genes. A total of 115 annotated genes, two miRNAs and three snoRNA were located within the eight ROH islands. The two ROH islands on ECA3, located between positions 35,477,778 and 37,590,699 bp, overlapped with four known QTL regions; two related to insect bite hypersensitivity (IBH) [49,50], one to guttural pouch tympany [51] and one to hair pigmentation [52]. The ROH island on ECA3: 106,769,095: 107,709,841 bp coincided with a known QTL for body height at the withers [53]. The ROH island on ECA11 coincided with four QTL regions related to insect bite hypersensitivity [49], hair density [54], overall body size [55] and height at the withers [56]. The ROH islands on ECA10, ECA15 and ECA19 did not overlap with any known QTLs.

## 4. Discussion

The evaluation of genetic diversity with traditional methods based on pedigree analyses have been widely used to manage animals’ genetic resources. In small livestock populations, where financial support is generally a limiting factor, pedigree-based methods are still commonly adopted due to their cost–effectiveness ratio [5,57,58,59,60]. However, with novel molecular and bioinformatics approaches, genetic variability can be more proficiently evaluated by using genomic information, and could lead to more precise and effective conservation programs. This study can be considered as the first step in deepening our knowledge about Bardigiano genomic diversity.

Effective population size (N_e_), computed using genealogical data, could be biased if pedigree depth and quality are low [61]. To this end, molecular markers could be a valuable alternative form of information for the characterization of N_e_ in livestock populations [62]. The value of N_e_ in the Bardigiano horses based on LD information was approximately 39 horses in the last generation. This value of N_e_ is lower than that calculated in the Persian Arabian Horses (N_e_ = 113) [63], in the Saxon Thuringa Coldblood (N_e_ = 48.1) and in the Rhenish German Draught Horse (N_e_ = 46.1) [64]. Interestingly, the N_e_ estimate based on pedigree data (calculated as individual increases in inbreeding [65,66] in the live Bardigiano population) was lower than (N_e_ = 30.7), yet comparable to, that found by molecular data analyses [5]. This evidence highlights that the N_e_ estimate obtained from pedigree can be considered a proper approximation of molecular based N_e_ in the case of Bardigiano horses. The estimation of N_e_ in the last 20 generations showed a steady decrease in effective population size, in line with the known history of the breed. The N_e_S analysis highlighted two sharp reductions in N_e_ in the Bardigiano breed, approximately ten and six generations ago. Since the average generation interval in this breed is 8.74 years [5], we hypothesized that those two reductions occurred, roughly, around 1933–1942 and 1968–1977, respectively. During World War II, the number of Bardigiano horses decreased dramatically, with only five stallions and 150 mares surviving [3]. This evidence may justify the sharp reduction in the N_e_ about 10 generations ago. The Bardigiano studbook was founded in 1977 with the aim to recover the breed, just after the second sharp reduction in N_e_ found in this study. This second sharp reduction in N_e_ might highlight the loss of genetic diversity due to the selection of breeding candidates during the initial steps of the breed recovery. In contrast, the general increase and stabilization of the N_e_ from the 4th generation onwards might reflect the breeding policies applied by the Bardigiano studbook to conserve the breed.

The extent and frequency of ROH have been widely used to infer ancestry at the individual and breed level [16]. Long ROH are generally considered to be a sign of recent inbreeding, whereas short ROH can capture ancient inbreeding which derived from older ancestors and can capture population bottlenecks. Under the assumption that 1 cM equals 1.24 Mbp, ROH can be separated in length classes to express different points in time when inbreeding occurred [19]. In the Bardigiano horse, the majority of ROH belonged to the shortest length class, being equal to 0.5–1 Mbp (69.5%). Similar results were observed in local breeds, as in the Bosnian Mountain Horse (61.7%) [32] and in the Noriker horse breed, where the majority (60–85%) of ROH segments were shorter than 4 Mbp [34]. This result might mainly originate from the small population size of the Bardigiano breed and the strong founder effect due to the experienced bottlenecks. In contrast, the lowest percentage of ROH (1.2%) resided into the longest class (ROH > 8Mbp), which highlights a rather small reduction in genetic variability occurred in the last generations. This result is coherent with the breeding strategies applied by the studbook to reduce matings between highly related animals. Likewise, the F_ROH_ ranged from 17.0%, considering all ROH regardless of the size, to 2.0%, considering only the longest class (ROH > 8Mbp). This last result highlights the occurrence of older inbreeding, rather than inbreeding which happened in the last few generations. Nevertheless, the average F_ROH_ in the Bardigiano was higher compared to other small-sized horse populations. For example, in the Croatian Posavje horse breed, the F_ROH_ was equal to 8.59%; in the Bosnian mountain horse (originating from Bosnia and Herzegovina) this value reached a mean of 13.21%; and in the Austrian Noriker horse it ranged between 8.00% and 13.00%, depending on coat color strains [31,34]. Similar levels of F_ROH_ to those in the Bardigiano horses were found in the Polish Koniks breed (mean = 16.00%), which is a primitive breed closely related to the extinct wild Tarpan [67], and in Purebred Arabian horses (mean =1 7.70%), despite the larger population size [32]. In contrast, a higher level of inbreeding was found in Friesian horses (mean = 22.30%) compared to the Bardigiano [68]. As expected from previous studies [7,69,70], the inbreeding based on molecular data in the Bardigiano tended to be higher than that previously found at the pedigree level [5].

In general, the ROHs caused by inbreeding tend to be distributed unevenly over the genome, as we have found in the Bardigiano horse, with a different distribution of numbers and sizes of ROH for each chromosome [71]. However, ROH that are located in specific genomic regions and shared among several individuals are thought to be potential signs of selection [24,25,27,71]. The reasons behind this theory are the driving forces of directional selection, which increase homozygosity around a target locus (ROH island). The breeding objective of the Bardigiano studbook can be divided into three main goals: (a) ensure long-term survival of the breed; (b) preserve the Bardigiano’s distinctive morphological features; and (c) increase the use of this breed in riding and draft activities [2]. For the latter aspect, in the ’90s, the Bardigiano studbook introduced a breeding value for the height at the withers to help the conversion of taller horses, which are thus more suitable for leisure. It is noteworthy that, in this study, we found ROH islands in two of the four loci previously found to explain 83% of the size variation in horses [55]. Specifically, one ROH was located on ECA3 and shared among 93% of the Bardigiano horses, which overlapped with the ligand dependent nuclear receptor corepressor-like (*LCORL*) gene [72]. The *LCORL* gene is a transcription factor that has been associated with human height [73,74,75]. In cattle, *LCORL* was identified in a screen for loci under selection [76], and the immediately adjacent gene, *NCAPG*, has been implicated in prenatal growth [77] as well as in the body size in Franches–Montagnes horses [77]. The other ROH that overlapped with a locus among the four explaining 83% of the size variation in horses was on ECA11, where the LIM and SH3 protein 1 (*LASP1)* gene is located. The *LASP1* gene mediates cell migration and survival, and its expression is induced by insulin-like growth factor (*IGF1*) [78]. However, the ROH on ECA11 is located within a gene-dense region counting thirty-four genes; therefore, we cannot exclude that selection in this region might target other genes. Based on the overlap with known QTLs for body size, we hypothesize that the ongoing selection toward higher horses might have led to a reduction in variability in those two regions.

In addition, this latter region of ECA11, together with two ROH islands on ECA3 between 35.48 and 37.59 Mbp, overlapped with known QTLs for equine insect bite hypersensitivity (IBH). IBH is a pruritic skin allergy caused primarily by biting midges, Culicoides spp [49]. This skin disease has polygenic inheritance and occurs at high frequencies in several horse breeds worldwide, thus causing increased costs and reduced horse welfare. To the best of our knowledge, so far, no cases of IBH have been reported in the Bardigiano breed, even if the animals are mostly farmed outside, and pasture in the mountain area is commonly practiced in the breed. The summer pasture exposure and the surrounding vegetation have been identified as risk factors associated with IBH in warmblood horses [79]. We therefore hypothesize that long and extensive exposure to insect bites might have shaped the genome of Bardigiano horses towards more resistant animals. Nevertheless, those three ROH islands located on ECA3 and ECA11 also overlapped with known QTLs for hair pigmentation [52] and hair density [54]. Interestingly, some of the distinctive features of the Bardigiano horse are actually related to hair type and pigmentation. Indeed, the only coat color accepted by the breeding association is bay, with a slight preference for dark bay, whereas chestnuts and extremely light bays are not allowed in the breed. In addition, only limited white markings on the legs and face are allowed and, in general, if present, they are scored as a defect in the horse evaluation [2]. In the study by Haase et al. 2013 [52], seven QTLs were significantly associated with the white marking phenotype, which explained ~54% of the total genetic variance [52,80]. The ROH island identified on ECA3 (36.13–37.59 Mbp) in the Bardigiano horses overlapped with the melanocortin receptor gene (*MC1R)*. The *MC1R* encoded by the Extension (E) locus controls, together with the peptide antagonist agouti-signaling-protein (*ASIP*), the amounts of melanin pigments in mammals [81]. We, therefore, cannot rule out the hypothesis that this region might be under selection pressure because of morphological requirements in terms of coat color in the Bardigiano breed.

## 5. Conclusions

The genetic diversity of the Bardigiano breed was analyzed via N_e_ based on LD and ROHs detected in this study. We identified a decrease in the LD-based N_e_, when calculated over the last 20 generations, in line with the known history of the breed. The ROH analysis provided additional information to further optimize the breeding and management decisions needed to ensure the long term survival of the breed. Inbreeding levels, especially those based on short ROH segments, were moderate to high, suggesting the presence of past bottlenecks in the genome of Bardigiano horses. Conversely, the lowest percentage of ROH resided in the longest ROH classes, showing a small reduction in genetic variability during the last generations. Eight ROH islands were found in the breed; these islands were shared among more than 70% of the Bardigiano horses analyzed in this study. Four of them overlapped with known QTLs associated with conformation traits and disease susceptibility. Genes related with conformation traits, coat color and disease susceptibility appear to be targets of directional selection in the Bardigiano breed. This study has outlined the first genome-wide map of selection signatures in an Italian native horse breed, and provides material for further functional studies on the biological mechanisms of complex traits in horses.

## Figures and Tables

**Figure 1 animals-10-01005-f001:**
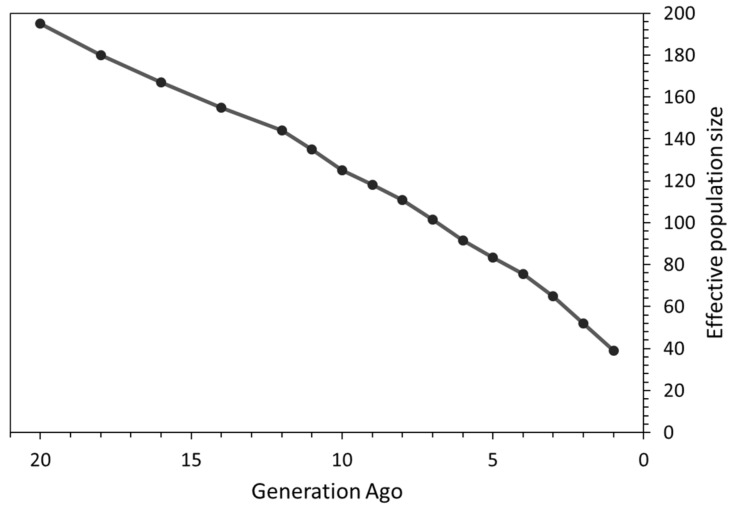
Plot of effective population size change between 20 generations ago and the last generation in the Bardigiano horse breed.

**Figure 2 animals-10-01005-f002:**
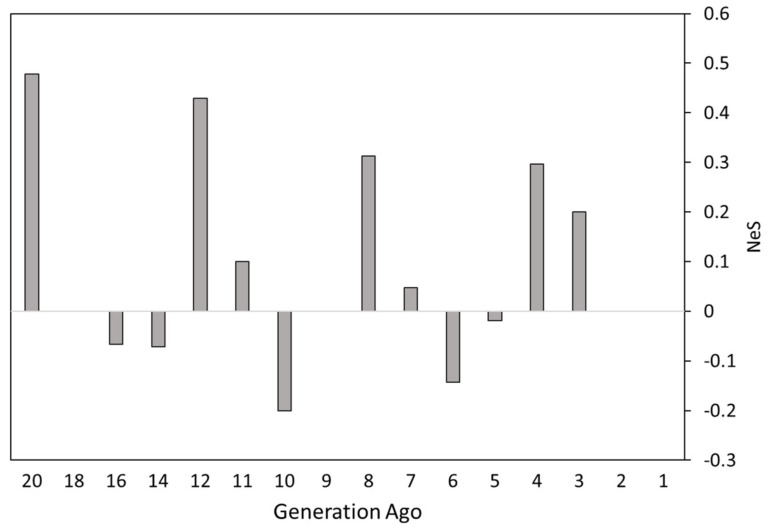
N_e_ Slope analysis (N_e_S) between 20 generations ago and the last generation in the Bardigiano horse breed. A constant rate of change in the N_e_ is shown as a flat line at 0 in the Y-axis.

**Figure 3 animals-10-01005-f003:**
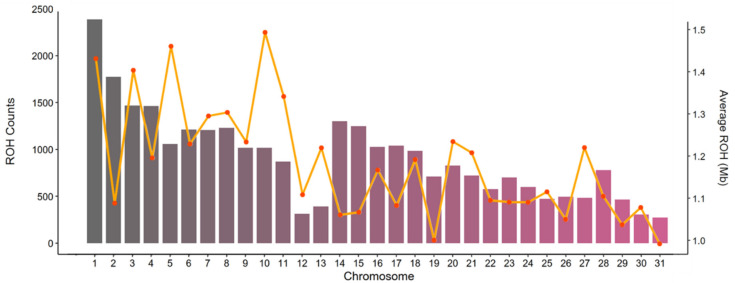
Distribution and average length of runs of homozygosity (ROH) in Mbp detected across the autosomal genome in the Bardigiano horses. The bar plots show the ROH counts per chromosome and the orange line shows the average ROH size (Mbp) per chromosome.

**Figure 4 animals-10-01005-f004:**
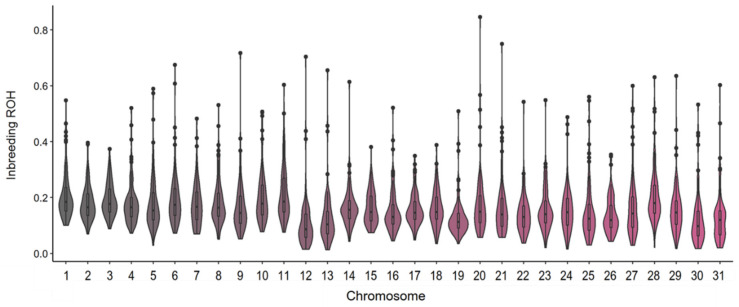
Violin plot of the inbreeding based on runs of homozygosity (F_ROH_) calculated per chromosome in the Bardigiano horse breed.

**Figure 5 animals-10-01005-f005:**
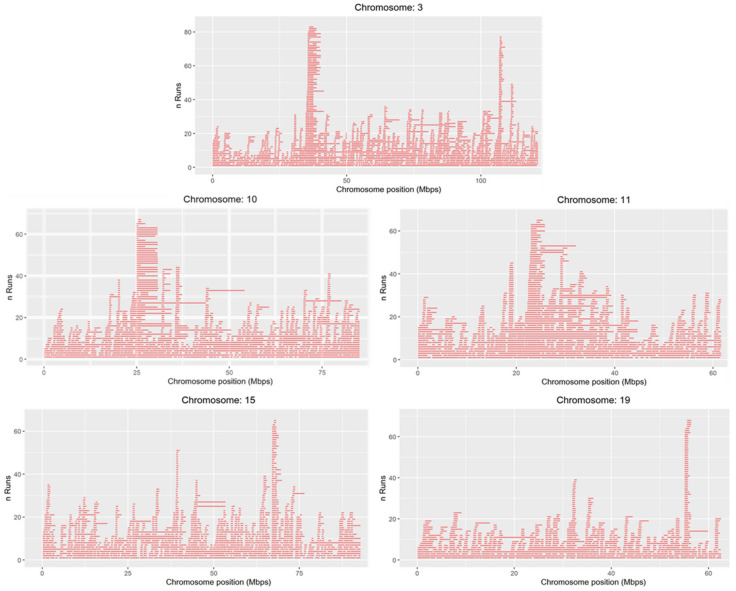
ROH islands on ECA3, ECA10, ECA11, ECA15 and ECA19 in the Bardigiano horse breed. The x-axis represents the chromosome position in Mbp and the y-axis represents the number of horses showing an ROH in each chromosome position.

**Table 1 animals-10-01005-t001:** Descriptive statistics of runs of homozygosity (ROH).

		ROH			
Length Class (Mbp)	N. ^1^	N_ROH_ ^2^	Percentage ^3^	S_ROH_ ^4^	L_ROH_ ^5^
0.5–1	88	19,746	69.5%	224.4	0.70
1–2	88	6073	21.4%	69.0	1.33
2–4	88	1587	5.6%	18.0	2.74
4–8	88	664	2.3%	7.5	5.42
>8	84	353	1.2%	4.2	13.40

^1^ Number of animals, ^2^ total number of ROH, ^3^ relative percentage, ^4^ average ROH number per animal, ^5^ average length of ROH.

**Table 2 animals-10-01005-t002:** Descriptive statistics of inbreeding based on runs of homozygosity (F_ROH_) within each ROH length class for the 88 Bardigiano horses that passed the QC.

	Inbreeding Based on ROH (F_ROH_)			
Length Class (Mbp)	Mean	Min. ^1^	Max. ^2^	SD ^3^
0.5–1	0.07	0.06	0.08	0.004
1–2	0.04	0.03	0.05	0.005
2–4	0.02	0.01	0.04	0.006
4–8	0.02	0.002	0.04	0.008
>8	0.02	0.00	0.09	0.02
Total	0.17	0.12	0.26	0.03

^1^ Minimum, ^2^ maximum, ^3^ standard deviation.

**Table 3 animals-10-01005-t003:** Runs of homozygosity (ROH) islands shared in over 70% of the Bardigiano horses, with genomic coordinates and a list of annotated genes located within each ROH island.

ECA ^1^	Start (bp) ^2^	End (bp) ^2^	Length (Kb)	Annotated Genes	% of Horses
3	35,477,778	36,012,699	535	*ZNF469, ZFPM1,ZC3H18, IL17C, CYBA, MVD, SNAI3, RNF166, CTU2, PIEZO1*	93%
3	36,131,080	37,590,699	1460	*CBFA2T3, ACSF3, CDH15, SLC22A31, ANKRD11, SPG7, RPL13, CPNE7, DPEP1, CHMP1A, SPATA33, CDK10, SPATA2L, VPS9D1, ZNF276, FANCA, SPIRE2, TCF25,* *MC1R,TUBB3, DEF8, DBNDD1, GAS8, PRDM7, CENPE, BDH2, SLC9B2*	92%
3	106,769,095	107,709,841	941	*LCORL, NCAPG, DCAF16, FAM184B,*	92%
10	24,965,648	25,501,456	536	*IL11, TMEM190, TMEM238, RPL28, UBE2S, SHISA7, ISOC2, C19orf85, ZNF628, NAT14, SSC5D, SBK2, SBK3, ZNF579, FIZ1, ZNF524, ZNF865, ZNF784, ZNF580, ZNF581, CCDC106, ZNF835, U2AF2, EPN1, RFPL4A, EQUCABV1R902, EQUCABV1R903, NLRP4, NLRP13, NLRP5*	76%
10	25,636,022	26,106,403	470	*EDDM13, ZNF667, ZNF583, ZNF582, SMIM17, ZNF471, ZFP28, ZNF470, ZNF71*	76%
11	22,957,922	24,177,108	1219	*CDK12, MED1, STAC2, CACNB1, ARL5C, PLXDC1, FBXO47, LINC00672, LASP1, RPL23, C17orf98, CWC25, PIP4K2B, PSMB3, MLLT6, PCGF2, CISD3, EPOP, SRCIN1, ARHGAP23, SOCS7, GPR179, MRPL45, NPEPPS, KPNB1, TBKBP1, TBX21, OSBPL7, MRPL10, LRRC46, SCRN2, SP6, SP2, PNPO*	73%
15	67,045,471	67,539,463	494	*LCLAT1, LBH, YPEL5*	73%
19	55,113,101	55,782,316	669	*//*	77%

^1^ ECA: Equine chromosome, ^2^ position in base pairs (bp) on EquCab3.

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
