# Peer review of "Genetic Diversity and Signatures of Selection in a Native Italian Horse Breed Based on SNP Data"

_animals, 2020, doi:10.3390/ani10061005_

Round 1
Reviewer 1 Report
The authors of “Genetic diversity and signatures of selection in a 2 Native Italian Horse breed based on SNP data” evaluated several population genetic metrics for the Bardigiano breed in order to better understand the current genetic diversity scenario and the effect of past bottleneck events on those metrics. The manuscript is very well-written and the and both analysis and results scientifically sound. There are some minor comments and suggestions which I will list below.
Lines 60-67: These sentences could be removed from the introduction in order to shorten this section.
Line 110: How many horses were used?
Line 114: 1.24 cM/Mb?? Centimorgan is a unit of recombination, which informs the minimal distance between two recombination spots. Probably the authors would like to inform that the cM units in this study was set as 1.24 Mb.
Line 165: What is the average generation interval for the breed? The authors discussed very well the historical events resulting in bottlenecks for the breed. Therefore, in order to analyze in deep, the effective population size fluctuations across the time, the number of generations can be matched with the average generation time. Consequently, the authors can try to find possible historical events responsible for these changes in the Ne.
Line 191: The authors could compare the average inbreed in each respective generation and match these results with the Ne results. The authors can estimate the expected ROH length, generated by inbreed events, in g generations ago using the following formula: 1/2g Morgans. For example, a ROH segment generated by an inbreed event which occurred 20 generations ago will have an expected length of 0.025 Morgans. It is important to highlight that this value is in Morgans, not centiMorgans. Additionally, the authors must use the previous determined relationship between Morgans and Mb (1.24) in order to estimate the expected ROH length. The authors can find more details about this analysis on:
de Souza Fonseca, P. A., dos Santos, F. C., Rosse, I. C., Ventura, R. V., Brunelli, F. Â. T., Penna, V. M., ... & Peixoto, M. G. C. D. (2016). Retelling the recent evolution of genetic diversity for Guzerá: Inferences from LD decay, runs of homozygosity and Ne over the generations. Livestock science, 193, 110-117.
Ferenčaković, M., Sölkner, J., & Curik, I. (2013). Estimating autozygosity from high-throughput information: effects of SNP density and genotyping errors. Genetics Selection Evolution, 45(1), 42.
Howrigan, D.P., Simonson, M.A., Keller, M.C., 2011. Detecting autozygosity through runs of homozygosity: a comparison of three autozygosity detection algorithms. BMC genomics 12, 1.
Line 201: The number of animals information of Table 2 is not necessary once all classes have the same number for animals.
Line 213: Check the font.
Line 227: Inform the exact position, not the rounded.
Line 250: Add space between “(“ and the reference.
Line 265: The authors reported 1cM equal to 1.24 MB previously.
Lines: 347-349: Talk about further applications in management for the breed regarding the results observed in the present study.
Tables: Centralize the titles.
Author Response
We would like to thank the reviewer for the review and the constructive raised suggestions. Please find attached our point by point answers to your comments.

Reviewer 2 Report
It is a very typical genetic diversity analysis on a domesticated animal breed. It is interesting and will serve a good genetic resource.
Author Response
We would like to thank the reviewer for the review and the positive evaluation of our work.
Reviewer 3 Report
My congratulations with the Authors for this well written paper. Methods sound and english language is fine.
I only have some minor comments.
----------------------------------------------------------------------------------
Please consider to rephrase odd expressions like:
"have paved the way" [60]
"did not overlapped" [229] with "did not overlap"
Check again the entire manuscript for such error/typo.
Table 3: I would suggest to simply use the universally recognised "chr" instead of "ECA" and the note...
Please rephrase:
The Ne estimation based ... Rhenish German Draught Horse (Ne=46.1) [247-250]
Interesting -> Interestingly [250]
This result is in line -> This result is coherent [273]
Please consider to rephrase the sentence from 306 to 308.
Mega basepair should be shorten as Mbp throughout the entire manuscript.
The English language in the conclusion section appear to be less sharp than in other part of the manuscript in terms of readability and punctuation. Please try to amend this part.
Author Response
We thank the reviewers for their constructive review. Please find attached our responses to the points raised. Changes in the manuscript are highlighted in yellow. We have found few more typos that are highlighted in light blue.
